# VDX-111 targets proliferative pathways in canine cancer cell lines

Kristen B. Farrell [1]*, Sunetra Das[1], Steven K. Nordeen[2], James R. Lambert[2], Douglas H. Thamm[1]

1 Department of Clinical Sciences, Flint Animal Cancer Center, Colorado State University, Fort Collins, CO, United States of America, 2 Department of Pathology, University of Colorado Anschutz Medical Campus, Aurora, CO, United States of America

* Kristen.farrell@colostate.edu

**Data Availability Statement:** All relevant data are within the manuscript and its Supporting Information files.

**Funding:** The author(s) received no specific funding for this work.

## Abstract

VDX-111 (also identified as AMPI-109) is a vitamin D derivative which has shown anticancer activity. To further assess the function of this compound against multiple cancer types, we examined the efficacy of VDX-111 against a panel of 30 well characterized canine cancer cell lines. Across a variety of cancer types, VDX-111 induced widely variable growth inhibition, cell death, and migration inhibition, at concentrations ranging from 10 nM to 1 µM. Growth inhibition sensitivity did not correlate strongly with tumor cell histotype; however, it was significantly correlated with the expression of genes in multiple cell signaling pathways, including the MAPK and PI3K-AKT pathways. We confirmed inhibition of these signaling pathways as likely participants in the effects of VDX-111. These results suggest that a subset of canine tumors may be sensitive to treatment with VDX-111, and suggests possible predictive markers of drug sensitivity and pharmacodynamic biomarkers of drug exposure that could be employed in future clinical trials.

## Introduction

$1\alpha,25$-dihydroxyvitamin $D_3$ ($1,25(OH)_2D_3$), the dihydroxylated metabolite of vitamin D3, has demonstrated antiproliferative properties on cancer cells. However, its potential as a therapeutic agent is limited due to its calcemic properties [1–3]. Thus, $1,25(OH)_2D_3$ derivatives that combine anti-tumoral properties with a lower calcemic activity may be more useful as anticancer therapies, either as single agents or in combination [4–6]. VDX-111 (also identified as AMPI-109) is a novel analog of $1,25(OH)_2D_3$ with an increased half-life through covalent binding to the vitamin D receptor ligand binding domain, thus reducing toxicity compared to the unmodified compound while increasing anti-proliferative effects [7]. VDX-111 has demonstrated efficacy against renal and triple-negative breast cancer through reduction of migration and invasion capacities and promoting apoptosis [8–11].

The PI3K-AKT pathway promotes survival and proliferation in response to extracellular signals and is commonly dysregulated in cancer cells. Receptor tyrosine kinases initiate the signaling cascade which involves activation of RAS, PI3K, and AKT, and numerous downstream targets. Activating mutations or amplifications of PI3KCA are common in some canine and

**Competing interests:** The authors have declared that no competing interests exist.

human cancer types, while the central signaling protein AKT is rarely mutated [12–16]. The MAPK pathway is also involved in cell proliferation, among other roles such as inflammation, metabolism, and motility, and other functions. ERK1/2 is an effector protein in the MAPK pathway, which when activated via phosphorylation directly and indirectly activates multiple transcription factors [17,18]. Both the PI3K-AKT and MAPK pathways have been shown to be influenced by $1,25(OH)_2D_3$ [5,19,20].

Canine cancers are pathologically indistinguishable from their human counterparts and display similar patterns of disease progression as well as similar mutational and genomic landscapes [21–25]. Canine cancer patients act as a unique opportunity to test novel therapeutics in spontaneously arising cancers before initiation of full clinical trials in humans; thus, establishing *in vitro* evidence of antitumor effects in canine cancer cell lines adds valuable preclinical data to justify canine clinical investigations.

In this study, we examined the efficacy of VDX-111 against a panel of well characterized canine cancer cell lines. A large number of cell lines demonstrated sensitivity to VDX-111 treatment, manifested by growth inhibition, induction of cell death, and reduced migration capabilities. Sensitivity in canine cell lines significantly correlated with expression of multiple cell regulatory pathways identified via gene expression analysis, including those involved in the MAPK and PI3K-AKT pathway among others. We confirmed VDX-111's targeting of these pathways by demonstrating inhibition of AKT and ERK1/2 phosphorylation.

## Materials and methods

### Cell culture and viability assays

A collection of 30 canine tumor-derived cell lines was authenticated by species specific PCR and STR analysis as described [26] and tested for mycoplasma before experiments. Adherent cells were grown in Dulbecco's modified Eagle medium (DMEM) and non-adherent cells were grown in RPMI-1640, both supplemented with 10% fetal bovine serum, MEM Vitamin solution, 2 mM L-glutamine, 1 mM sodium pyruvate, non-essential amino acids, and antibiotic/antimycotic solution (Corning, Corning, NY). All cells were grown in a 37°C humidified atmosphere of 5% $CO_2$ and serially passaged by trypsinization or density gradient centrifugation. Drug sensitivity assays were performed by plating 2000 cells/well in a 96-well plate, allowing cells to adhere overnight if necessary, followed by addition of varying concentrations of VDX-111 and a 72-hour incubation. Resazurin (MP Biomedicals, Santa Ana, CA) 200 μg/mL was added at 10% volume followed by 1.5-hour incubation at 37°C, then fluorescence was used to quantify live cells using a Bio-Tek Synergy HT Multi-Mode Microplate Reader. Relative viable cell number was expressed as a percentage of cells treated with ethanol in equal volume as the highest drug concentration. Five wells were measured per condition and the experiment was repeated two to three times. Results from independent experiments were combined into a total of 10–15 replicates per data point.

### Live cell imaging

Canine cell lines were transfected with Nuclight Red™ (Essen Bioscience, Ann Arbor, MI), selected with 1 μg/mL puromycin, plated at 1000 cells/well in a 96-well plate and allowed to adhere overnight. Media was replaced with media containing VDX-111 or ethanol control and the cell death indicator YOYO™-1 Iodide (491/509) (Invitrogen, Carlsbad, CA) at 100 nM. Images were obtained every 6 hours after drug addition for up to 96 hours using an IncuCyte Zoom live cell imager (Sartorius, Edgewood, NY). IncuCyte Zoom software was then used for fluorescent object threshold selection and object counting. Four images were collected per well and averaged. Five wells per condition were further averaged to create each data point.

## Migration assays

Cells were serum starved overnight in media containing 0.1% FBS before plating 30,000 cells/well in the top chamber of Boyden chambers (8.0 μm pores) (Corning, Corning, NY) in a 24-well plate. Complete media containing 10% FBS filled the lower well outside of the chamber. Drug treatment or ethanol control was added to the top chamber containing cells, and two replicates of each condition were included. Wells without a chamber but containing drug were included to account for cell death and quantified using resazurin as described above. Cells were allowed to incubate in chambers for 18 hours. Non-migrated cells were then removed from the top of the membrane using a cotton swab and migrated cells on the membrane stained with Diff-Quik. Membranes were then dried and mounted on slides for brightfield imaging. Cell numbers from five 10x fields of each membrane (10 total per condition) were quantified using ImageJ (NIH, Bethesda, MD), and percent cell death measured in a simultaneous alamar blue assay was accounted for in the final totals.

## Gene expression correlation with drug sensitivity

RNA was extracted from canine cell lines using RNeasy kit (Qiagen, Germantown, MD) following manufacturer's protocol. RNA was quantified on a NanoDrop Microvolume Spectrophotometer and quality was assessed by TapeStation or Bioanalyzer (Agilent, Santa Clara, CA). The total RNA was processed using Universal Plus™mRNA-Seq library preparation kits with NuQuant® and the resulting cDNA library was sequenced on Illumina NovaSeq6000 to generate 150 bp paired end reads. Illumina RNAseq fastq files were processed to remove low quality reads and adaptors using Trimmomatic (v.0.36) [27]. The canine paired end reads were mapped against CanFam3.1 (ensembl v99) genome, respectively, using STAR aligner (v2.6.1a) [28]. The raw count data was generated using—quantMode option within STAR aligner.

The raw count data was filtered to remove genes with read count <4 in at least 4 samples and was normalized using the DESeq2 method [29]. The expression of each gene was correlated with the percent survival at 1 μM or 100 nM of the corresponding cell line (listed in **Fig 1**) using Pearson correlation method; multiple testing correction was done by using the FDRtool R package [30]. The genes with correlation coefficients associated with ap-value of <0.05 were used for downstream pathway analysis. The significantly correlated genes were functionally annotated and enriched pathways were identified via Metascape [31] and gProfiler tools [32]. The pathways with FDR of <0.05 were considered enriched.

## Differential expression and gene set enrichment analysis

Differential expression analysis between sensitive (n = 3) and resistant (n = 5) cell lines was done using DESeq2 [33]. To identify differentially expressed genes (DEGs), the raw counts for all genes with low counts were eliminated and genes with at least 10 reads in at least 3 samples were retained. Following normalization of counts, the DEGs were called using DESeq function and genes with fold change of log2 >1.5 and adjusted p-value <0.05 were defined as differentially expressed genes in sensitive cell lines when compared to resistant cell lines. Additionally, Gene Set Enrichment Analysis (GSEA) was used to identify enriched MSIGDB gene sets (using Collections: Hallmark and Canonical Pathways) in sensitive and resistant groups [34]. The false discovery rate (FDR) <0.25 was used to identify significantly enriched gene sets in sensitive and resistant cell lines.

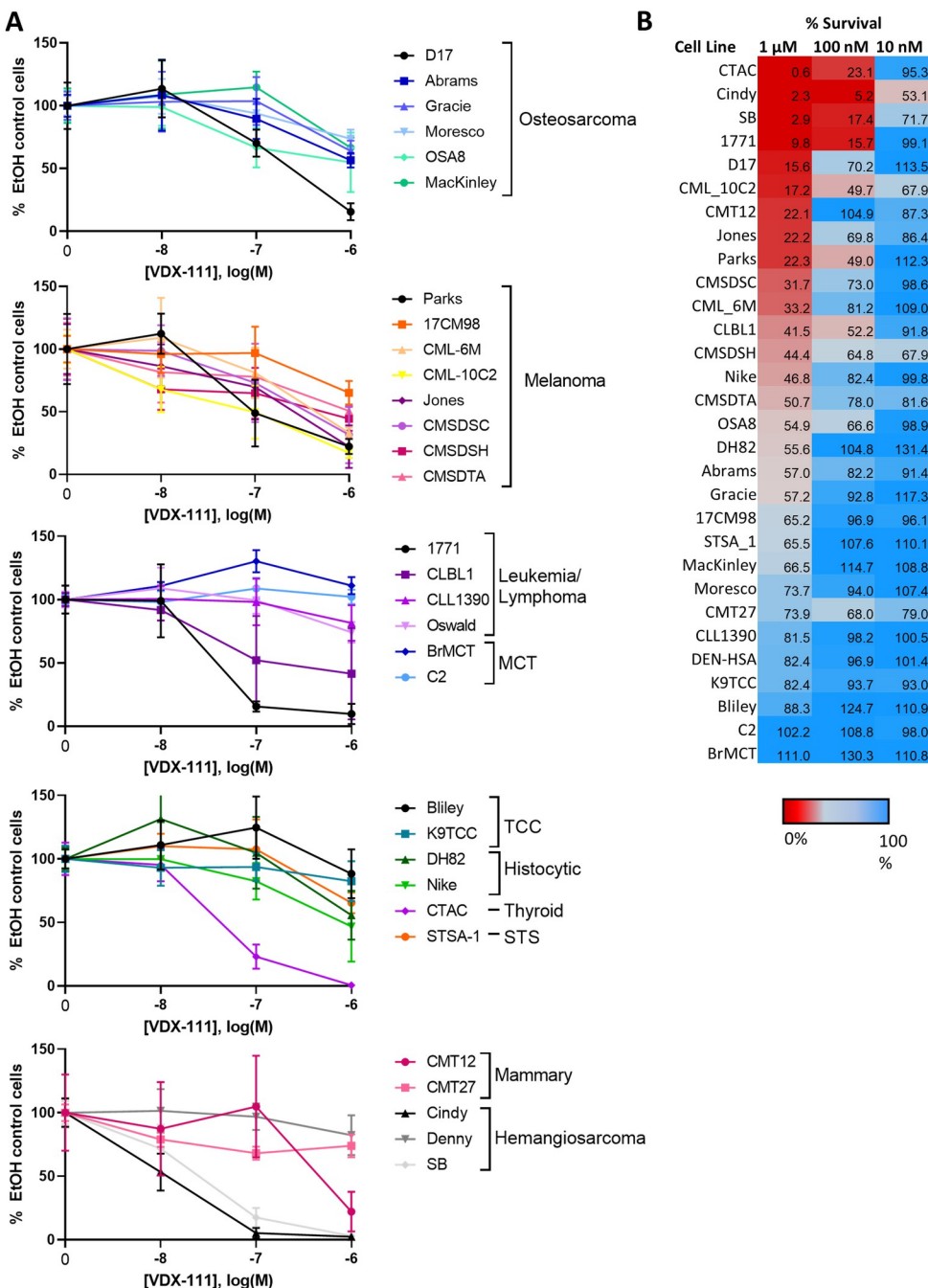

**Fig 1. Sensitivity of canine cell lines to VDX-111.** (A) Survival of canine cell lines after 72 hours of VDX-111 exposure. MCT = mast cell tumor, STS = soft tissue sarcoma. Cell lines of the FACC Canine Cell Line panel cover a variety of cancer types and display a variety of sensitivities at 1 μM and 100 nM VDX-111. Error bars represent SD. (B) Heat map displaying the percent of cells surviving compared to ethanol control after 72 hours in VDX-111 exposure, as measured via resazurin fluorescence.

## Western blots

After 4 hours of VDX-111 exposure, cells were lysed on the plate with cell lysis buffer [7 mL M-Per (Thermo Scientific, Waltham, MA), 140 mg sodium dodecyl sulfate (Fisher Chemical, Waltham, MA), 1/2 Complete Mini Protease Inhibitor Cocktail tablet (Roche, Indianapolis,

IN), 1 mM NaOVA (Sigma, St. Louis, MO), 1 mM PMSF (Sigma, St. Louis, MO)] on ice for 5 minutes. Samples were then aspirated using a 25G needle 5x, centrifuged at 23000 x g for 15 minutes at 4°C, and supernatants collected. A BCA assay (Biorad, Hercules, CA) was performed to quantify protein concentrations of each sample. Equal protein amounts were then boiled for 5 min with SDS sample buffer. Proteins were separated using SDS-PAGE on a NuPAGE 4–12% Bis-Tris protein gel (Invitrogen, Waltham, MA) followed by transfer onto PVDF membranes (Immobilon- Milipore, Burlington, MA) and blocking with Superblock T20 (TBS) Blocking Buffer (Thermo Scientific, Waltham, MA). Primary antibodies [anti-AKT (Cell Signaling #9272), anti-P-AKT (Cell Signaling #4060), anti-ERK (Cell Signaling #4695), anti-P-ERK (Cell Signaling #4370), cofilin (Cell Signaling #5175) (Cell Signaling, Danvers, MA)] were used at 1:1000 dilutions in blocking buffer at 4°C overnight, and secondary antibody at 1:40,000 (Goat anti-rabbit IgG HRP-conjugated (Thermo Scientific Waltham, MA)) for 1 hour at room temperature. Membranes were imaged with chemiluminescent substrate (Supersignal West Pico, Thermo Scientific Waltham, MA) using a Chemi Doc XES+ imaging system (BioRad, Hercules, CA). Membranes were first blotted and imaged with the phospho-antibody, then stripped using Restore PLUS Western Blot Stripping Buffer (Thermo Scientific, Waltham, MA), followed by repeating the method from the blocking step with the corresponding un-phosphorylated protein primary antibody. Quantification was performed using ImageJ (NIH, Bethesda, MD) by first normalizing all protein amounts to the cofilin loading control, then comparing relative amounts of phosphorylated and total protein.

## Cell cycle analysis

Cells were plated and allowed to attach overnight before replacing media with media containing 100 nM VDX-111, 1 μM VDX-111, or equivalent volume of ethanol. After 72 hours of treatment, $1 \times 10^6$ cells were fixed with ice cold 70% ethanol, washed with PBS, and resuspended in 500 μL FXCycle PI/RNase Staining Solution (Molecular Probes, Eugene, OR) for 15 minutes. Cells were then assessed using a Beckman Coulter Gallios flow cytometer with the 488 nm laser and 620/30 nm filter. Results were visualized using FlowJo (BD, Franklin Lakes, NJ) software by first selecting singlets based on forward and side scatter, then creating a histogram with gates around areas of interest.

## Statistics

Graphs were generated and statistical analysis (student's t-test, one-way ANOVA, two-way ANOVA) were performed using GraphPad Prism 9 (GraphPad Software, Boston, MA). P-values less than 0.05 are indicated with a single asterisk (*), less than 0.01 with two asterisks (**), less than 0.001 with three asterisks (***), and less than 0.0001 with four asterisks (****).

## Results

### Canine cell lines are variably sensitive to VDX-111

The FACC canine cell line panel [35] was tested for sensitivity of canine cancer cell lines to growth inhibition by VDX-111. **Fig 1** displays the survival of each cell line compared to an ethanol control after 72 hours of exposure at 3 different concentrations of VDX-111. Cell lines varied dramatically in their sensitivity to VDX-111, and there was no obvious correlation with cancer histotype or driver mutation [21]. Three cell lines with differing sensitivities were selected for further study: Bliley, a bladder transitional cell carcinoma with relatively low sensitivity to VDX-111 (IC50 1.2 μM), DH82, a histiocytic sarcoma with moderate sensitivity to VDX-111 (IC50 359 nM), and CTAC, a thyroid carcinoma with high sensitivity to VDX-111 (IC50 74 nM).

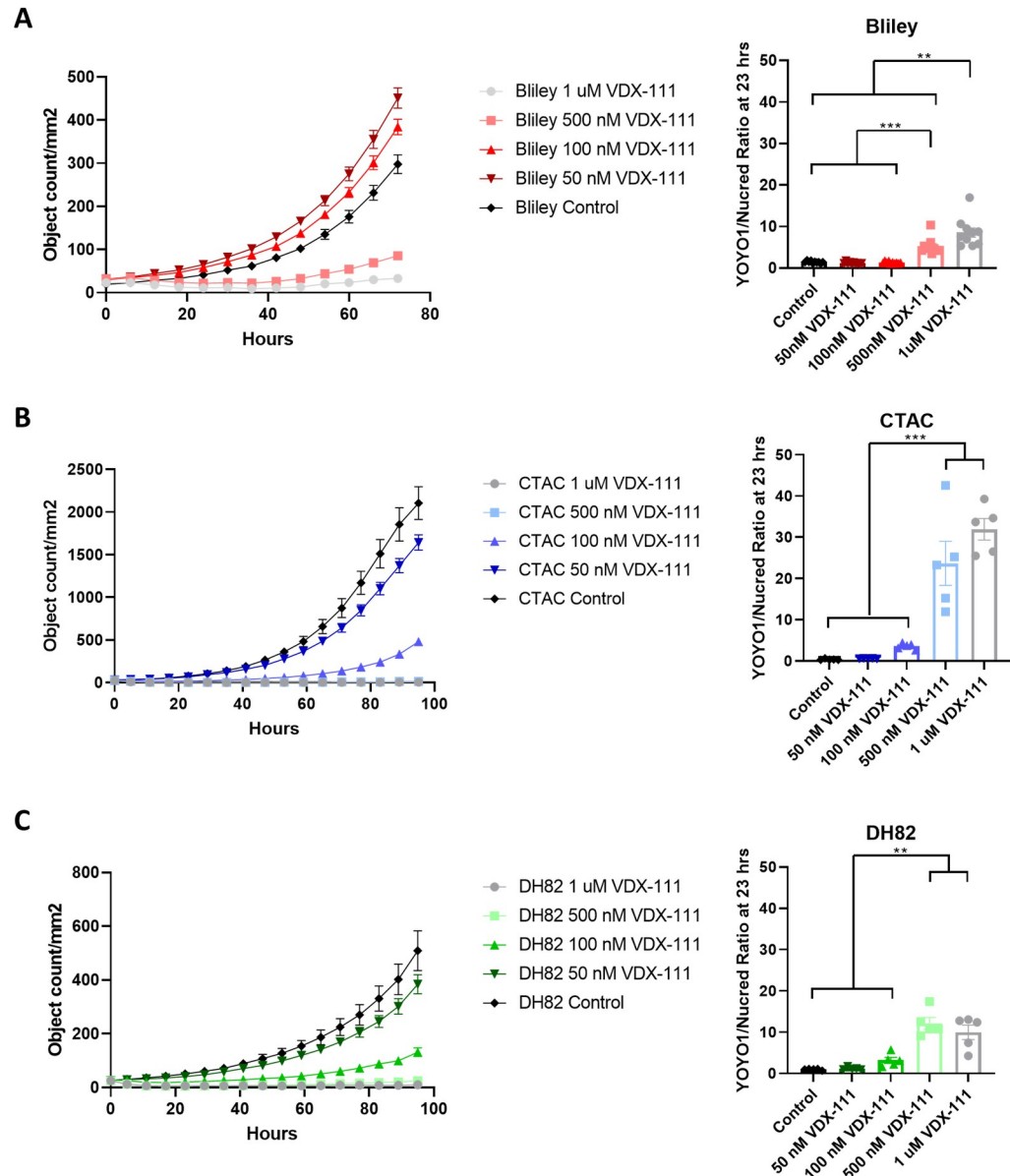

**Fig 2. VDX-111 induces apoptosis.** (A) Bliley cells expressing NucLight Red were imaged over time under VDX-111 and YOYO-1 exposure. An object count of red fluorescence indicates robust Bliley cell growth under low concentrations and some growth even under the highest VDX-111 concentrations (left). A green fluorescence object count, compared to total red cells indicates a low but present rate of apoptosis at 23 hours of drug exposure (right). (B) CTAC cells expressing NucLight Red were imaged over time under VDX-111 and YOYO-1 exposure. An object count of red fluorescence shows CTAC sensitivity to high VDX-111 concentrations. All groups are statistically different by two-way ANOVA (****$p<0.0001$) except 1 uM compared to 500 nM (left). A green fluorescence object count, compared to total red cells indicates high rates of apoptosis at 23 hours of drug exposure (right). (C) DH82 cells expressing NucLight Red were imaged over time under VDX-111 and YOYO-1 exposure. An object count of red fluorescence shows DH82 sensitivity to high VDX-111 concentrations. All groups are statistically different by two-way ANOVA (****$p<0.0001$) (left). A green fluorescence object count, compared to total red cells indicates moderate rates of apoptosis at 23 hours of drug exposure (right). Error bars represent SEM. ***$<0.001$ **$<0.01$.

Live cell imaging of these three cell lines when treated with VDX-111 confirmed inhibition of growth, and demonstrated induction of cell death via YOYO1 fluorescence (**Fig 2 and S1–S3 Videos**). A direct object count results in slight variation from the concentrations shown in

Fig 1 which are measured indirectly by cell reduction capacity, but similar patterns of sensitivity are displayed. Bliley, a less sensitive cell line, showed growth kinetics similar to control even at a VDX-111 concentration of 100 nM, and at a high concentration of 1 μM induced minimal apoptosis with a YOYO1/Nuclight Red ratio of less than 10. The growth of CTAC, the most sensitive cell line, was significantly inhibited at all concentrations over 50 nM VDX-111, and high cell death rates were observed at both 500 nM and 1 μM with a YOYO1/Nuclight Red ratio of over 20. DH82 also displayed little cell division over 50 nM, but exhibited moderate cell death rates at 500 nM and 1 μM. At 100 nM in both CTAC and DH82, there were low rates of cell growth but also little apoptosis, suggesting that VDX-111 causes additional growth inhibition beyond induction of apoptosis. However, no substantial changes in cell cycle were observed after 72 hours of treatment, suggesting that a stage-specific cell cycle block is not a significant mechanism of action (**S1 Fig in S1 File**).

The same three cell lines treated with VDX-111 in Boyden chamber assays displayed reductions in migration, again suggesting a mechanism beyond simply cell death (**Fig 3**). Cells were first serum starved overnight before treating with VDX-111, then and allowed to migrate towards serum-containing media for 18 hours across a membrane containing 8.0 μm pores. The reduction in ability to migrate also correlated with growth inhibition sensitivity, with Bliley demonstrating the smallest reduction in migration and CTAC the greatest. A simultaneous alamar blue assay quantifies cell growth during this time frame, and quantified migrated cells were corrected for any growth inhibition in treated vs. control cells. A dose dependent effect was observed with a greater reduction in migration with 1 μM treatment compared to 100 nM, even with correction for growth inhibition. These data suggest a pleiotropic mechanism of action of VDX-111 on canine tumor cells.

## Sensitivity to VDX-111 correlates with gene expression of proliferative pathways

The percent survival of all cell lines at both 1 μM and 100 nM were correlated with previous RNAseq data from the FACC panel using a correlation matrix. All findings are displayed in **S1 Table**, while genes with a correlation above +/- 0.6 are displayed in **Fig 4**. A positive correlation indicates greater survival with higher expression, while a negative correlation indicates lower survival with higher expression. Interestingly, we found some of the highest positive and negative correlating genes were signaling pathway effectors (MAX, SETD9). Correlation graphs showing gene expression and growth inhibition compared to control (as shown in **Fig 1**) are displayed, with the three further studied cell lines identified in color (**Fig 4B**). Interestingly, expression of the phosphatases PTPN3 and PTP4A3, previously identified as putative drug targets [10,11], had no correlation with sensitivity in the canine cell lines suggesting alternative targets in canine and triple-negative breast cancer cells (**S2 Fig in S1 File**).

We further performed pathway analysis on the significantly correlated genes using Metascape and g:profiler (**Fig 5 and S3-S5 Figs in S1 File**). An input of both positively and negatively correlated genes ($p<0.05$), resulted in a list of 1261 (1 μM) or 700 (100 nM) for the analysis. Many diverse cellular processes were returned, with both concentrations of VDX-111 demonstrating similar pathways (**S3 and S4 Fig in S1 File**). Intriguingly, many of the top correlating pathways were related to chromatin organization and histone modification. Of interest to our lab were numerous signaling pathways that were revealed as significantly enriched in the correlated genes, including the MAPK and PI3K-AKT pathways ($p<1E-6$ and $p<0.01$ respectively) (**Fig 5A**). Heat maps displaying the differing levels of expression of genes in these pathways identified using the Metascape pathway analysis show variations in expression between sensitive and resistant cell lines (**Fig 5B and S5 Fig in S1 File**).

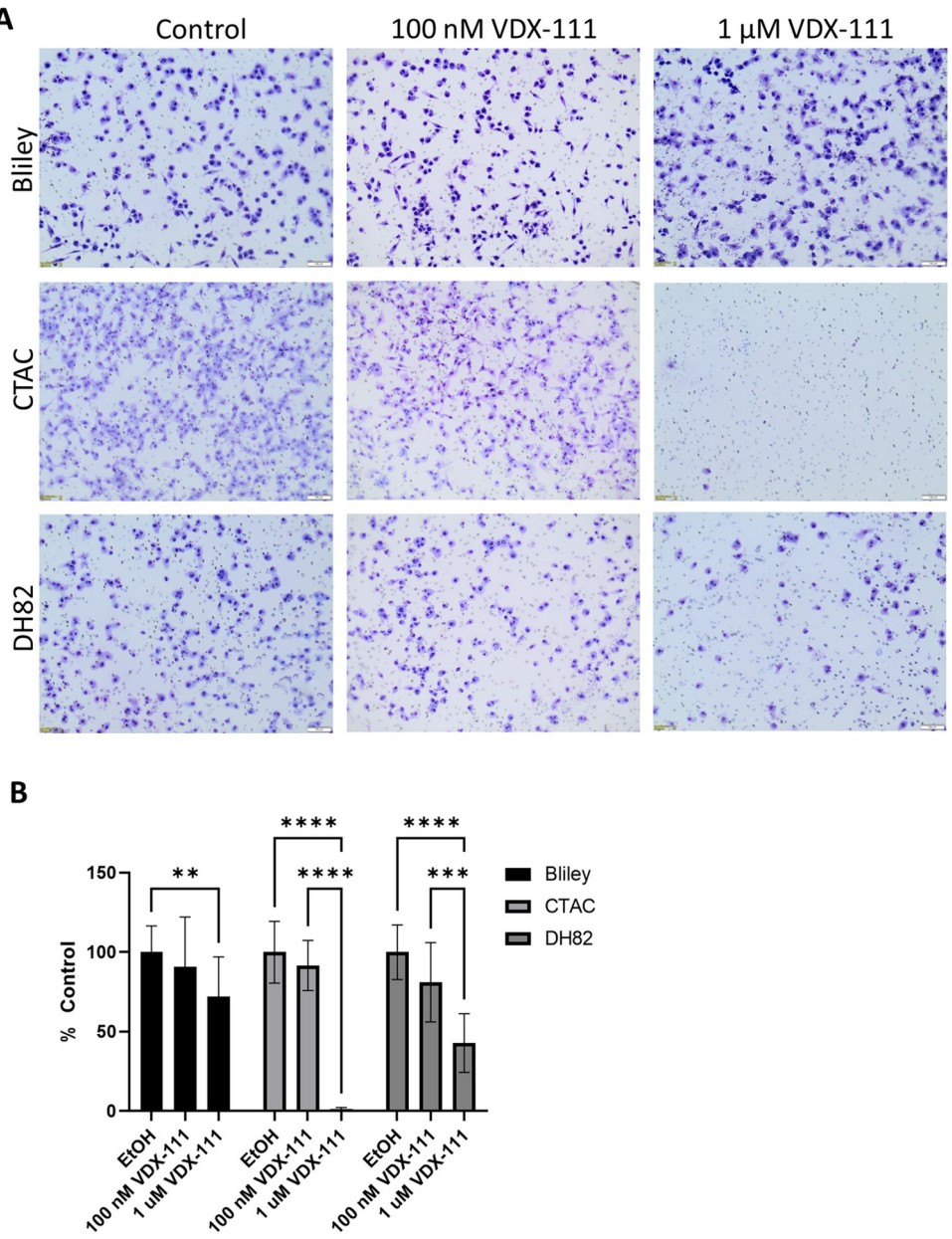

**Fig 3. VDX-111 inhibits migration.** (A) Cells were allowed to migrate across a membrane with 8.0 μm pores for 18 hours with or without VDX-111. Cells were then stained, imaged, and counted. (B) Number of migrated cells on the membrane per imaged 10x field of view were normalized to the control condition. Ten fields were quantified for each condition. Wells without membrane were also quantified via resazurin fluorescence to account for any cell death and reduced proliferation in the 18-hour period, and the percent difference in live cells was removed from the control counts. Error bars represent SD. ****<0.0001 ***<0.001 **<0.01.

Differentially expressed genes were also evaluated between the most sensitive and resistant cells lines. RNAseq data from sensitive cells (CTAC, Cindy, SB) were compared to resistant cells (CLL1390, DEN-HSA, Bliley, C2, BrMCT) as a pairwise comparison using DESeq2, and differentially expressed genes with adjusted p-value or FDR 1.5 & <(-1.5) are shown in **S2 Table**. Additionally, GSEA was performed using normalized expression of 12,383 genes in these sensitive vs. resistant cells. Numerous pathways uncovered likely have to do with the

**A** Genes with a survival and expression correlation of (+/-)0.6 or higher with 1 µM

| Ensembl gene ID | Gene Name | Pearson Correlation Coefficient 1 µM | p-value 1 µM | Pearson Correlation Coefficient 100 nM | p-value 100 nM |
|---|---|---|---|---|---|
| ENSCAFG00000000309 | FUCA2 | 0.660 | 0.000 | 0.455 | 0.019 |
| ENSCAFG00000016212 | MAX | 0.658 | 0.000 | 0.392 | 0.048 |
| ENSCAFG00000030661 | TTC33 | 0.649 | 0.000 | 0.449 | 0.022 |
| ENSCAFG00000013305 | DNAJC12 | 0.634 | 0.001 | 0.375 | 0.059 |
| ENSCAFG00000017268 | SLU7 | 0.624 | 0.001 | 0.433 | 0.027 |
| ENSCAFG00000003430 | UFL1 | 0.622 | 0.001 | 0.439 | 0.025 |
| ENSCAFG00000018338 | CLUL1 | 0.617 | 0.001 | 0.391 | 0.048 |
| ENSCAFG00000017121 | SLC51B | 0.615 | 0.001 | 0.466 | 0.016 |
| ENSCAFG00000031981 | ZAR1L | 0.615 | 0.001 | 0.418 | 0.034 |
| ENSCAFG00000005534 | MBD5 | 0.614 | 0.001 | 0.425 | 0.030 |
| ENSCAFG00000012941 | TFRC | 0.612 | 0.001 | 0.429 | 0.029 |
| ENSCAFG00000032746 | MXD1 | 0.608 | 0.001 | 0.398 | 0.044 |
| ENSCAFG00000003389 | THRAP3 | 0.608 | 0.001 | 0.450 | 0.021 |
| ENSCAFG00000020175 | PBX3 | 0.606 | 0.001 | 0.407 | 0.039 |
| ENSCAFG00000007423 | CPEB3 | 0.603 | 0.001 | 0.455 | 0.019 |
| ENSCAFG00000004938 | UNC13D | 0.602 | 0.001 | 0.404 | 0.041 |
| ENSCAFG00000047118 | N/A | 0.600 | 0.001 | 0.391 | 0.048 |
| ENSCAFG00000000588 | ALDH7A1 | -0.610 | 0.001 | -0.524 | 0.006 |
| ENSCAFG00000000214 | CCDC112 | -0.617 | 0.001 | -0.588 | 0.002 |
| ENSCAFG00000014075 | MB21D2 | -0.637 | 0.000 | -0.488 | 0.011 |
| ENSCAFG00000006065 | PRNP | -0.664 | 0.000 | -0.458 | 0.019 |
| ENSCAFG00000006922 | SETD9 | -0.672 | 0.000 | -0.508 | 0.008 |
| ENSCAFG00000043673 | N/A | -0.674 | 0.000 | -0.701 | 0.000 |
| ENSCAFG00000013084 | N/A | -0.696 | 0.000 | -0.482 | 0.013 |
| ENSCAFG00000009150 | OGFOD1 | -0.735 | 0.000 | -0.623 | 0.001 |

**B**

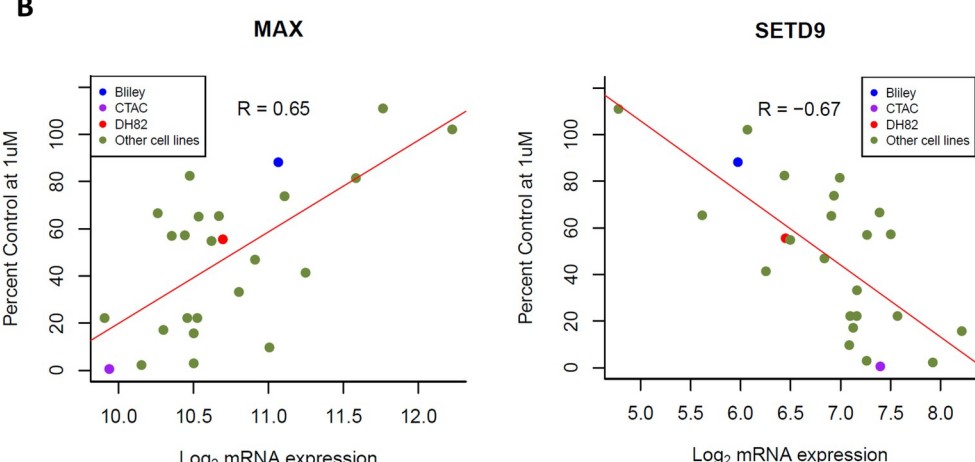

**Fig 4. Canine cell line gene expression correlation with VDX-111 sensitivity identifies components of the PI3K-AKT pathway.** (A) An expression matrix was created correlating gene expression measured by RNAseq with the percent survival of the canine cells under VDX-111 exposure. Displayed on the chart are the genes with correlation (+/-) 0.6 with 1 uM treatment, showing the correlation at 1 µM, 100 nM, the p-values of each and gene name. A positive correlation indicates greater survival with higher expression, while a negative correlation indicates lower survival with higher expression. (B) Graphs showing the FACC canine cell line panel expression of MAX and SETD9 correlating to survival at 1 µM.

differing histotypes of the limited numbers of cell lines used in the pairwise comparison. However, resistant cells were found to be enriched in the KIT signaling/PI3K pathway, and MAPK pathway (**S6 Fig in S1 File**). Specific genes identified in each gene set for the sensitive and resistant cells are listed in **S3 Table**.

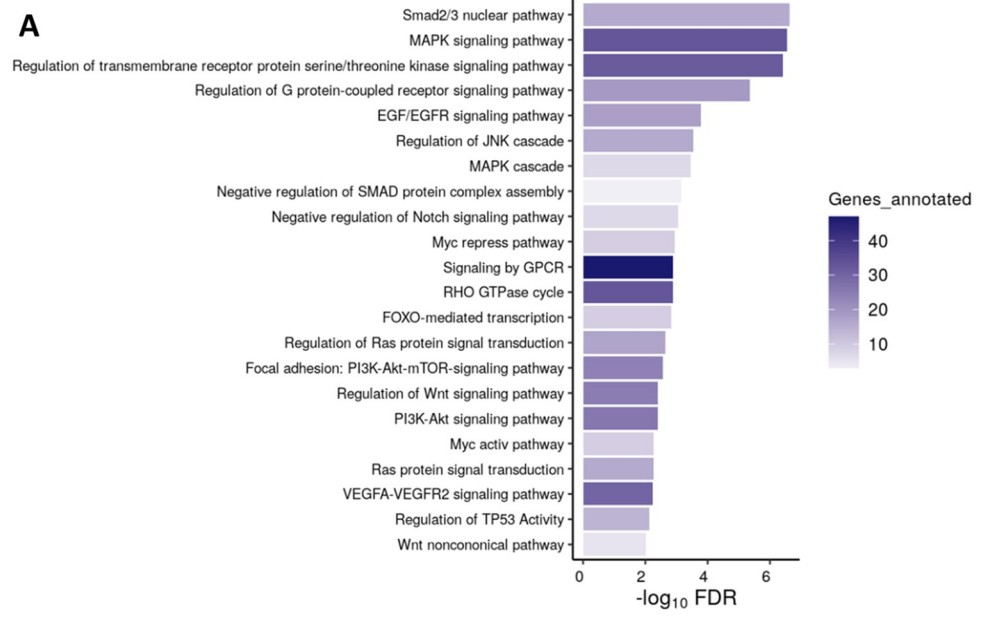

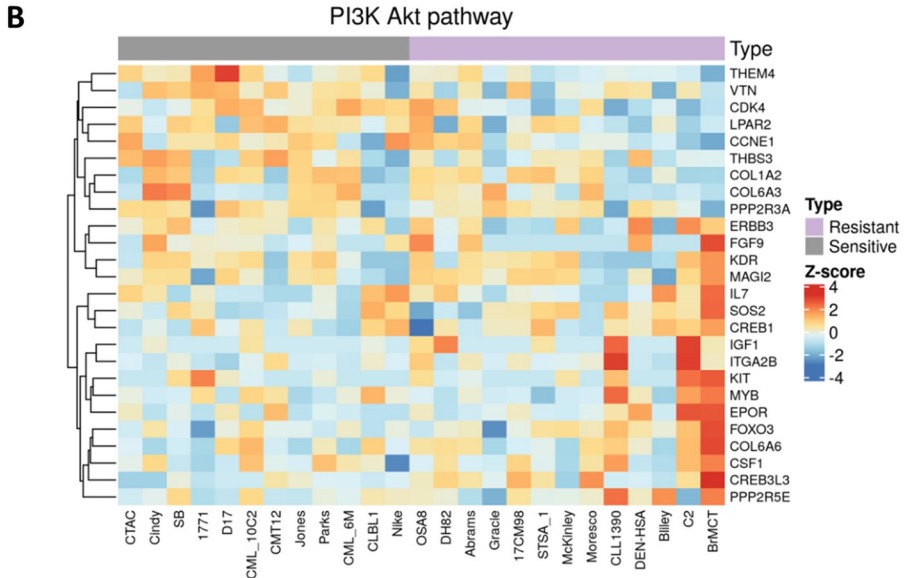

**Fig 5. Pathway analysis of gene expression correlating with VDX-111 survival.** (A) Pathway analysis using Metascape (metascape.org) with an input of 1261 significantly correlated genes (p<0.05) including either a positive or negative correlation with 1 μM treatment. Multiple signaling pathways were identified among the significantly correlated genes. (B) Heatmap of expression levels in each cell line of genes in the Metascape analysis of the PI3K-AKT pathway. Cell lines with >50% survival at 1 μM treatment are indicated as resistant, <50% survival at 1 μM treatment are indicated as sensitive.

We confirmed involvement of the MAPK and PI3K-AKT pathways with western blot analysis. Cell lines of varying sensitivities to VDX-111 showed dose-dependent changes in phosphorylation of AKT and ERK1/2 (**Fig 6**). The three previously utilized cell lines, along with Parks (melanoma–sensitive), 17CM98 (melanoma—moderate) and C2 (mast cell tumor–resistant), were assessed for AKT and ERK1/2 phosphorylation after 4 hours of exposure to VDX-111. The sensitive CTAC cells showed little change in phosphorylation, although this may be

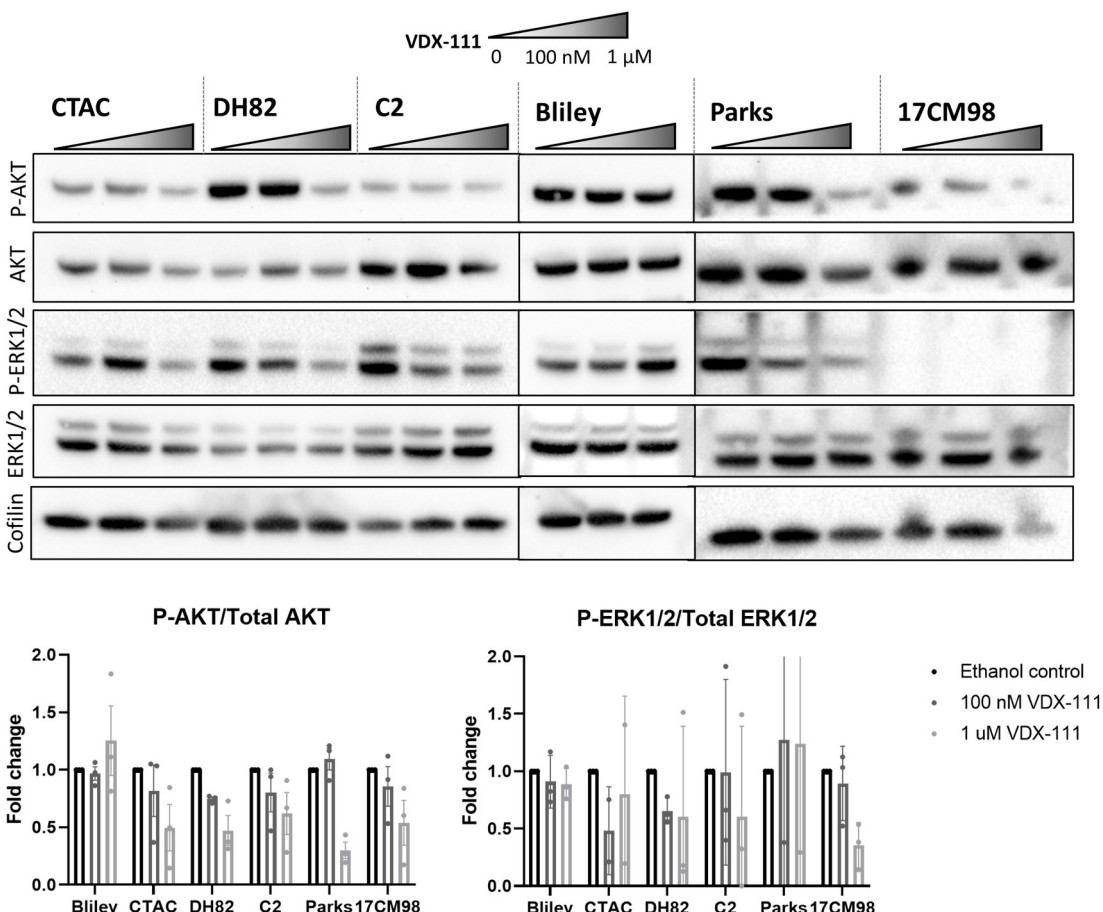

**Fig 6. Phosphorylation of growth promoting pathways is reduced with VDX-111 treatment.** Cells were treated with ethanol control, 100 nM, or 1 μM VDX-111 for 4 hours before harvesting lysates for western blot. Images were quantified with ImageJ and corrected to cofilin levels. Ratios were then normalized to the control condition. Three separate experiments were quantified to obtain averages shown in graphs. Error bars represent SD.

due to cell death already occurring with the extreme sensitivity these cells show to the drug (after only 5 hours of drug treatment, the Incucyte data described in **Fig 2** show a YOYO-1 to total cell ratio of 13:1 at 1 μM and 2:1 at 100 nM, indicating high levels of apoptosis after a short exposure for the CTAC cell line).

Another sensitive cell line, Parks, showed a dose dependent decrease in phosphorylation of both AKT and ERK1/2. The moderately sensitive cell line DH82 also showed a dose dependent decrease in phosphorylation of both AKT and ERK1/2, and the moderately sensitive cell line 17CM98 showed decreases phosphorylated AKT and ERK1/2. The resistant cell line Bliley showed a slight decrease in phosphorylated AKT, but no change in the ERK1/2 phosphorylation. The resistant cell line C2 showed no change in phosphorylated AKT but a dose dependent decrease in phosphorylation of ERK1/2. Thus, even cell lines that were relatively resistant to the growth inhibition effects of VDX-111 showed signaling changes in at least one of the proliferative pathways after treatment.

## Discussion

VDX-111 caused substantial growth inhibition in many canine cancer cell lines. Efficacy varied across a panel of 30 cell lines, from very high sensitivity to no sensitivity at a 1 μM

concentration. Growth inhibition and apoptosis were confirmed with live cell imaging, and reduced migration capabilities were also demonstrated. Though there was no apparent correlation between cell line histotype and sensitivity, a correlation matrix comparing sensitivity in the growth inhibition assay to gene expression via RNAseq data from each cell line revealed significance in numerous cell signaling pathways, among other cell functions (**S4 Fig in S1 File** and **S1 Table**). ERK1/2 and AKT both showed reduced levels of phosphorylation in numerous cell lines via western blot, confirming changes in at least these two signaling pathways after treatment. This implies that inhibition of AKT and/or MAPK pathway signaling may be playing a role in the antineoplastic effects of VDX-111, but does not rule out the potential involvement of other important pathways. One of the transcripts with the highest correlation to resistance to VDX-111-mediated growth inhibition is MAX, the major binding partner of c-Myc. C-Myc and MAX together act as a transcription factor integrating multiple signaling mechanisms and regulating numerous cellular responses. C-Myc is commonly dysregulated in both human and canine tumors [36–40]. The correlation of MAX expression with survival suggests that overexpression of MAX may act as a resistance mechanism to VDX-111, and potentially other therapeutics that target aberrant cell signaling. The overrepresentation of the MAPK and PI3K pathways in the most resistant cells via pairwise analysis also suggests the importance of these signaling pathways and may indicate involvement in resistance, especially as these pathways have been demonstrated to regulate Myc function [41–43]. Other cell functions with significant correlation to expression, such as chromatin organization and histone modification, may warrant further investigation to further elucidate the mechanisms of this drug's efficacy. The importance of controlled access of regulatory proteins to the genome has been demonstrated in numerous cancers [44,45], and its interplay with the pathways in this study as well as the vitamin D receptor certainly may herald future important findings for VDX-111 [46–48].

Vitamin D has been studied for decades for its relation to cancer prevention and treatment. The anti-tumor properties of vitamin D and its analogs are mediated by vitamin D receptor (VDR) binding to vitamin D response elements in DNA and subsequent changes in gene expression [49,50]. Numerous analogs have been created in attempts to provide VDR binding while providing reduced toxicity to the patient after long exposure [6,50]. VDX-111 was designed for binding to the VDR-ligand binding domain to increase binding, half-life, and decrease catabolism. However, in triple-negative breast cancer models VDX-111 exerts its effects through a non-VDR-dependent mechanism demonstrating a lack of dependence on the VDR signaling axis[9]. Instead, the oncogenic phosphatase PRL-3 has been implicated as a major direct target of the drug [9–11]. This has been demonstrated in previous studies [7], as well as an unexpected mechanism as a PRL-3 phosphatase inhibitor [9–11]. In triple-negative breast cancer cell lines, treatment with VDX-111 resulted in a reduction in p-ERK1/2 similar to our canine cell lines, and this reduction in ERK activation was replicated with PRL-3 knockdown [11]. Notably, the siRNA screen which identified PRL-3 had other high-ranking hits from signaling pathways, such as EGFR [9]. The lack of correlation of anti-proliferative effects with PRL-3 expression we demonstrated in canine cancer cell lines does not rule out PRL-3 inhibition as a mechanism; it merely suggests that if this mechanism is functional in canine cells, PRL-3 may not play a role as an oncogenic driver in resistant cell and tumor models. Interestingly, we did identify another phosphatase, PTPN7, as a significant gene in both the cell panel-wide correlation matrix and the pairwise analysis between top sensitive and resistant cell lines (**S1** and **S2 Tables**). Our pathway association study suggests that multiple cell pathways may be related to the efficacy of this compound against cancer cells.

Regardless of the mechanism, effects of VDX-111 on the PI3K-AKT pathway have now been demonstrated in both human and canine cells. Thus, evaluation of the MAPK and PI3K-AKT pathway in patient samples may give insight into the potential for success with

VDX-111 as a therapeutic. Anti-proliferative effects of VDX-111 may be successful in combination with cytotoxic drugs or agents targeting additional signaling pathways. Further investigation into the other pathways identified in the correlation matrix may also reveal therapeutic targets for this compound and potential combination therapies.

## Supporting information

**S1 Raw images.**
(PDF)

**S1 File. S1 Fig: VDX-111 has minimal effect on cell cycle distribution.** A-C) Cells were exposed to VDX-111 or EtOH control for 72 hours before measuring PI fluorescence using flow cytometry. S2 Fig: VDX-111 sensitivity is not correlated with phosphatase expression despite established mechanism of action. A) Plots showing the FACC canine cell line panel PTPN3 expression correlating to survival at 1 μM and 100 nM. B) Plots showing the cell line panel PTP4A3 expression correlating to survival at 1 μM and 100 nM. S3 Fig: g:Profiler pathway analysis of gene expression correlating with VDX-111 sensitivity. Significantly correlated pathways for sensitivity at 1 μM. No pathways were significant for 100 nM. S4 Fig: Metascape pathway analysis of gene expression correlating with VDX-111 sensitivity. Significantly correlated pathways for sensitivity at 1 μM and 100 nM, inputting both positively and negatively correlated genes with a p>0.05. S5 Fig: Heatmap of expression levels in each cell line of genes in the "MAPK pathway". Expression levels are displayed of the genes identified in the Metascape analysis of the MAPK pathway. Cell lines with >50% survival at 1 μM treatment are indicated as resistant, <50% survival at 1 μM treatment are indicated as sensitive. S6 Fig: GSEA of the top sensitive and most resistant cell lines. GSEA using normalized expression of 12,383 genes where I have compared untreated cell lines that were sensitive (CTAC, Cindy, SB) and resistant (CLL1390, DEN-HSA, Bliley, C2, BrMCT) to VDX-111.
(PDF)

**S1 Table. Canine genes with expression correlated to VDX-111 survival.**
(CSV)

**S2 Table. Pairwise comparison of cell lines.** Sensitive (CTAC, Cindy, SB) and resistant (CLL1390, DEN-HSA, Bliley, C2, BrMCT) cell lines were compared using DESeq2, and differentially expressed genes with adjusted p-value or FDR 1.5 & <(-1.5).
(CSV)

**S3 Table. GSEA enriched gene sets in sensitive and resistant cell lines.**
(XLSX)

**S1 Video. Bliley (NucRed) + 100 nM VDX-111 and 100 nM YOYO1.**
(MP4)

**S2 Video. CTAC (NucRed) + 100 nM VDX-111 and 100 nM YOYO1.**
(MP4)

**S3 Video. DH82 (NucRed) + 100 nM VDX-111 and 100 nM YOYO1.**
(MP4)

## Acknowledgments

We thank Rupa Idate and Barbara Rose for their work in mycoplasma treatment, STR mapping, and RNAseq of the canine cell line panel.

## Author Contributions

**Conceptualization:** Kristen B. Farrell, Sunetra Das, Steven K. Nordeen, James R. Lambert, Douglas H. Thamm.

**Data curation:** Kristen B. Farrell, Sunetra Das.

**Formal analysis:** Kristen B. Farrell, Sunetra Das.

**Funding acquisition:** Douglas H. Thamm.

**Investigation:** Kristen B. Farrell, Sunetra Das.

**Methodology:** Kristen B. Farrell, Sunetra Das.

**Project administration:** Douglas H. Thamm.

**Resources:** Steven K. Nordeen, James R. Lambert, Douglas H. Thamm.

**Supervision:** Douglas H. Thamm.

**Visualization:** Kristen B. Farrell, Sunetra Das.

**Writing – original draft:** Kristen B. Farrell.

**Writing – review & editing:** Kristen B. Farrell, Sunetra Das, Steven K. Nordeen, James R. Lambert, Douglas H. Thamm.

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
