## [Decision Letter · Decision Letter 0]

4 Mar 2024

PONE-D-24-03607A novel small molecule targets proliferative pathways in canine cancer cell linesPLOS ONE

Dear Dr. Farrell,

Thank you for submitting your manuscript to PLOS ONE. After careful consideration, we feel that it has merit but does not fully meet PLOS ONE’s publication criteria as it currently stands. Therefore, we invite you to submit a revised version of the manuscript that addresses the points raised during the review process. Please submit your revised manuscript by Apr 18 2024 11:59PM. If you will need more time than this to complete your revisions, please reply to this message or contact the journal office at plosone@plos.org. Please include the following items when submitting your revised manuscript:A rebuttal letter that responds to each point raised by the academic editor and reviewer(s). You should upload this letter as a separate file labeled 'Response to Reviewers'.A marked-up copy of your manuscript that highlights changes made to the original version. You should upload this as a separate file labeled 'Revised Manuscript with Track Changes'.An unmarked version of your revised paper without tracked changes. You should upload this as a separate file labeled 'Manuscript'.

We look forward to receiving your revised manuscript.

Kind regards,

Dominique Heymann, Ph.D.

Academic Editor

PLOS ONE

Reviewers' comments:

Reviewer's Responses to Questions

**Comments to the Author**

1. Is the manuscript technically sound, and do the data support the conclusions?

Reviewer #1: Partly

Reviewer #2: Yes

2. Has the statistical analysis been performed appropriately and rigorously? 

Reviewer #1: N/A

Reviewer #2: Yes

3. Have the authors made all data underlying the findings in their manuscript fully available?

Reviewer #1: Yes

Reviewer #2: Yes

4. Is the manuscript presented in an intelligible fashion and written in standard English?

Reviewer #1: Yes

Reviewer #2: Yes

5. Review Comments to the Author

Reviewer #1: The manuscript by Farrell et al describes the effect of a novel small molecule – the vitamin D derivative VDX-111 – on a panel of 30 canine cancer cell lines to understand whether it could be further developed towards use in tumor therapy. The rationale that canine cancer presents a good and clinically accessible model also to further human cancer therapy is highly topical and of growing interest. Overall, the manuscript is clearly written, the sequence of experiments is logical and the quality of the laboratory work seems solid.

The paper starts with a not really surprising finding that the different cancer cell lines tested show a high heterogeneity with regards to reaction to the compound. The authors then assess the reason for cell growth inhibition in a few of the cell lines more in detail, showing that the compound induces a certain extent of apoptosis.

They then go on to assess whether the migratory capacity of sensitive cells is also impaired. This is where I don’t quite agree on the interpretation: if the CTAC cells are highly sensitive to VDX-111 (as they also suggest in the description to Figure 6, where it is mentioned that possibly already 5h of incubation might cause a lot of cell death), the reduction in cells seen after migration seems mostly to be an effect of cell death rather than specific decrease in migratory capacity. This might be checked also by assessing the number of cells that have migrated instead of counting the number of cells that have ‘disappeared’ from the original well. I think this would be important to address, should this claim be upheld. The same is true for DH82 cells, too.

With regards to Figure 4 it might be nice to show this data also as correlation matrix, not just as blank table. This would help interpretation. Also, what was the rationale to include the 3 additional canine cell lines here?

I do appreciate the enrichment of MAPK and PI3K-associated pathways in the correlated genes. However, the heatmap shown in Fig. 5B is not very striking; I actually doubt that unsupervised clustering would reveal any separation between resistant and sensitive cells in these targets; I think this should be assessed. Also, why not try to identify correlating genes or pathways using e.g. single sample GSEA or similar? As it stands, the majority of difference in the R vs S subpopulations seems to derive from 3 or 4 cell lines on the very right… Maybe restricting to less cell lines for R and S would help narrow down the aspect more precisely?

With regards to Figure 6, there is some nice data, but also some confusion. E.g. the text says “the moderately sensitive cell line 17CM98 showed decreases phosphorylated AKT and ERK1/2”. This is really not something I can support – firstly, the last lane is having some issues with the Cofilin loading control, which suggests that a decrease in signal is more a problem of decreased loading of the sample or Western blotting problems. Secondly, I cannot see a signal for P-ERK1/2 at all, so I am not sure how the conclusion that there is a decrease in this depending on the drug can be drawn. Also, I guess it should say ‘decreases in phoshporylated’, if I’m not mistaken. Also, the cell line Bliley seems to show – if anything – an increase in pERK signal, and I cannot really see the suggested decrease in pAKT either. Moreover, as there are differences in both pathways in the resistant as well as the sensitive cell lines, I am not sure whether the effect on these pathways is actually responsible for any of the survival defects observed; actually the data would argue the opposite. Finally, it might just be an overall ‘shock’ effect of being exposed to a drug in general that just leads to slight changes in these signaling pathways… I feel more mechanistic insight would be very important here before claiming that VDX-111 acts through PI3K and MAPK pathways. For instance, one could assess its effect on cell lines with a known PI3K or MAPK resistance to further corroborate this idea.

Finally, a minor comment: would it maybe make sense to mention the molecule in the title?

As such, while I find the study interesting, I think it needs a bit more work to really substantiate the claims put forward in the abstract.

Reviewer #2: The authors have conducted a comprehensive study investigating the effects of VDX-111, a vitamin D derivative, on a set of 30 well-characterized canine cancer cell lines. The study covers growth inhibition, cell death, migration inhibition, and gene expression of multiple cell signaling pathways at concentrations ranging from 10 nM to 1 μM. I find no faults with the study, which has been appropriately performed with well-designed and described methodology. I highly recommend its publication after minor modifications:

- The quality of the figures is low and sometimes makes reading difficult, especially for Figure 5.

- Regarding viability tests, why was the only studied incubation period 72 hours?

- Why was only one concentration of VDX-111 evaluated for the cell migration tests?

- In the "Results" section, "Sensitivity to VDX-111 correlates with gene expression of proliferative pathways," the titles and descriptions of Figures 5 and 6 should be separated. Additionally, the sentence "Future investigation into these pathways may reveal additional mechanisms of action of this compound" should be moved to the discussion section and expanded to provide more content to the discussion.

- In the "Discussion" section, I believe the first paragraph could be more developed and include more bibliographic references. The authors refer to MAX and c-Myc. Have any articles explored the overexpression of the transcription factors c-Myc and Max in certain canine tumors?

6. PLOS authors have the option to publish the peer review history of their article (what does this mean?). If published, this will include your full peer review and any attached files.

Reviewer #1: No

Reviewer #2: **Yes: **Benjamin Cartiaux

---

## [Author Response · Author response to Decision Letter 0]

3 Apr 2024

Kristen B. Farrell

Flint Animal Cancer Center

Department of Clinical Sciences

Colorado State University, Fort Collins, CO

Kristen.Farrell@colostate.edu

April 3, 2024

Dominique Heymann, Ph.D.

Academic Editor

PLOS ONE

Thank you to the editor and reviewers for their constructive comments on our manuscript “A novel small molecule targets proliferative pathways in canine cancer cell lines”. We have addressed the issues and clarifications suggested by the reviewers and are submitting an improved copy of the manuscript, with changes marked in red. Please see each reviewer comment addressed individually below. 

Sincerely,

Kristen Farrell

Reviewer #1: The manuscript by Farrell et al describes the effect of a novel small molecule – the vitamin D derivative VDX-111 – on a panel of 30 canine cancer cell lines to understand whether it could be further developed towards use in tumor therapy. The rationale that canine cancer presents a good and clinically accessible model also to further human cancer therapy is highly topical and of growing interest. Overall, the manuscript is clearly written, the sequence of experiments is logical and the quality of the laboratory work seems solid.

The paper starts with a not really surprising finding that the different cancer cell lines tested show a high heterogeneity with regards to reaction to the compound. The authors then assess the reason for cell growth inhibition in a few of the cell lines more in detail, showing that the compound induces a certain extent of apoptosis.

They then go on to assess whether the migratory capacity of sensitive cells is also impaired. This is where I don’t quite agree on the interpretation: if the CTAC cells are highly sensitive to VDX-111 (as they also suggest in the description to Figure 6, where it is mentioned that possibly already 5h of incubation might cause a lot of cell death), the reduction in cells seen after migration seems mostly to be an effect of cell death rather than specific decrease in migratory capacity. This might be checked also by assessing the number of cells that have migrated instead of counting the number of cells that have ‘disappeared’ from the original well. I think this would be important to address, should this claim be upheld. The same is true for DH82 cells, too.

Author response: We agree that cell death should be considered in the migration assay. We have already included mention that cell death/growth inhibition was considered by running a simultaneous Alamar blue assay quantifying metabolically active cells in each condition and the percent differences in treated wells were added back into the final migrated cell count. Language in the manuscript has been changed to clarify this. 

With regards to Figure 4 it might be nice to show this data also as correlation matrix, not just as blank table. This would help interpretation. Also, what was the rationale to include the 3 additional canine cell lines here?

Author response: We have added a visual correlation matrix for the genes listed in the table as the top results of the correlation matrix. The full results of the correlation are available in the supplement. All cell lines from figure 1 were included in the analysis, and the three cell lines used in the detailed studies were emphasized in the graphs. We hope this answers the reviewer’s question, and we are happy to make additional clarifications if needed.

I do appreciate the enrichment of MAPK and PI3K-associated pathways in the correlated genes. However, the heatmap shown in Fig. 5B is not very striking; I actually doubt that unsupervised clustering would reveal any separation between resistant and sensitive cells in these targets; I think this should be assessed. Also, why not try to identify correlating genes or pathways using e.g. single sample GSEA or similar? As it stands, the majority of difference in the R vs S subpopulations seems to derive from 3 or 4 cell lines on the very right… Maybe restricting to less cell lines for R and S would help narrow down the aspect more precisely?

Author response: We thank the reviewer for this suggestion and have performed the requested pairwise analysis and GSEA. The pairwise analysis was run with a “sensitive” and “resistant” group representing the top 3 most sensitive cell lines (CTAC, Cindy, SB) and 5 most resistant cell lines (CLL1390, DEN-HSA, Bliley, C2, BRMCT). The resulting significant differentially expressed genes are included in the supplemental data, as well as a GSEA comparing these two groups. Description of the analysis is included in the results and described in the methods, and we have added S6 Fig, S2 Table, and S3 Table. 

With regards to Figure 6, there is some nice data, but also some confusion. E.g. the text says “the moderately sensitive cell line 17CM98 showed decreases phosphorylated AKT and ERK1/2”. This is really not something I can support – firstly, the last lane is having some issues with the Cofilin loading control, which suggests that a decrease in signal is more a problem of decreased loading of the sample or Western blotting problems. Secondly, I cannot see a signal for P-ERK1/2 at all, so I am not sure how the conclusion that there is a decrease in this depending on the drug can be drawn. Also, I guess it should say ‘decreases in phoshporylated’, if I’m not mistaken. Also, the cell line Bliley seems to show – if anything – an increase in pERK signal, and I cannot really see the suggested decrease in pAKT either. Moreover, as there are differences in both pathways in the resistant as well as the sensitive cell lines, I am not sure whether the effect on these pathways is actually responsible for any of the survival defects observed; actually the data would argue the opposite. Finally, it might just be an overall ‘shock’ effect of being exposed to a drug in general that just leads to slight changes in these signaling pathways… I feel more mechanistic insight would be very important here before claiming that VDX-111 acts through PI3K and MAPK pathways. For instance, one could assess its effect on cell lines with a known PI3K or MAPK resistance to further corroborate this idea.

Author response: In the Figure 6 western blot, the cofilin loading control was included in calculations to account for any slight variations in loading. The density quantifications were first all normalized to the cofilin loading controls for each cell line. Among the cofilin bands, the control condition for each cell line was set as 1 and divergences in the other treatment lanes as % change from 1. The density quantifications of the other proteins in each lane were then multiplied by the % of change in cofilin to account for loading variations. Once density was adjusted to cofilin levels, only then were phospho-protein and total protein levels compared across treatments in each cell line. The western blot was repeated 3 times from separately performed biological replicates, as shown in the quantification as individual measurements, in order to account for variations in individual blots. Representative images were chosen but may not perfectly display the exact average change calculated by 3 repeated experiments. 

We agree that more mechanistic studies will be helpful in the future. In this study however, we are not attempting to describe the entire mechanism of action of the drug, but would simply like to suggest that these pathways are important (shown by the differentially expressed genes) and affected by (shown in the western blot) the drug treatment. We believe the bioinformatic data along with the physical protein data in the western blot combine to convey an involvement of these pathways. 

Finally, a minor comment: would it maybe make sense to mention the molecule in the title?

Author response: We have added the molecule name in the title. 

Reviewer #2: The authors have conducted a comprehensive study investigating the effects of VDX-111, a vitamin D derivative, on a set of 30 well-characterized canine cancer cell lines. The study covers growth inhibition, cell death, migration inhibition, and gene expression of multiple cell signaling pathways at concentrations ranging from 10 nM to 1 μM. I find no faults with the study, which has been appropriately performed with well-designed and described methodology. I highly recommend its publication after minor modifications:

- The quality of the figures is low and sometimes makes reading difficult, especially for Figure 5.

Author response: High quality images were uploaded, and we will ensure high quality images are uploaded again. 

- Regarding viability tests, why was the only studied incubation period 72 hours?

Author response: Cell viability is commonly measured after a 72-hour incubation period, as can be found in historical as well as recent studies using chemotherapeutic agents (1-2). Our lab performs all assays with a 72 hour incubation for consistency. 

1. Larsson, P., Engqvist, H., Biermann, J. et al. Optimization of cell viability assays to improve replicability and reproducibility of cancer drug sensitivity screens. Sci Rep 10, 5798 (2020).

2. Hafner M, Niepel M, Chung M, Sorger PK. Growth rate inhibition metrics correct for confounders in measuring sensitivity to cancer drugs. Nat Methods. 2016 Jun;13(6):521-7. 

- Why was only one concentration of VDX-111 evaluated for the cell migration tests?

Author response: We agree this could be expanded to demonstrate a dose-dependent effect. We have repeated the migration assays using an additional concentration of 100 nM. The results of figure 3 have been updated. 

- In the "Results" section, "Sensitivity to VDX-111 correlates with gene expression of proliferative pathways," the titles and descriptions of Figures 5 and 6 should be separated. Additionally, the sentence "Future investigation into these pathways may reveal additional mechanisms of action of this compound" should be moved to the discussion section and expanded to provide more content to the discussion.

Author response: The figure legends were separated and the sentence considering other pathways was moved to the discussion as suggested. 

- In the "Discussion" section, I believe the first paragraph could be more developed and include more bibliographic references. The authors refer to MAX and c-Myc. Have any articles explored the overexpression of the transcription factors c-Myc and Max in certain canine tumors?

Author response: Thank you for this suggestion, we have added references to the discussion about the expression of Myc in several canine tumors and expanded the discussion.

---

## [Decision Letter · Decision Letter 1]

16 Apr 2024

VDX-111 targets proliferative pathways in canine cancer cell lines

PONE-D-24-03607R1

Dear Dr. Farrell,

We’re pleased to inform you that your manuscript has been judged scientifically suitable for publication and will be formally accepted for publication once it meets all outstanding technical requirements.

Kind regards,

Dominique Heymann, Ph.D.

Academic Editor

PLOS ONE

Additional Editor Comments (optional):

Reviewers' comments:

Reviewer's Responses to Questions

**Comments to the Author**

1. If the authors have adequately addressed your comments raised in a previous round of review and you feel that this manuscript is now acceptable for publication, you may indicate that here to bypass the “Comments to the Author” section, enter your conflict of interest statement in the “Confidential to Editor” section, and submit your "Accept" recommendation.

Reviewer #1: All comments have been addressed

Reviewer #2: All comments have been addressed

2. Is the manuscript technically sound, and do the data support the conclusions?

Reviewer #1: Yes

Reviewer #2: Yes

3. Has the statistical analysis been performed appropriately and rigorously? 

Reviewer #1: Yes

Reviewer #2: Yes

4. Have the authors made all data underlying the findings in their manuscript fully available?

Reviewer #1: Yes

Reviewer #2: Yes

5. Is the manuscript presented in an intelligible fashion and written in standard English?

Reviewer #1: Yes

Reviewer #2: Yes

6. Review Comments to the Author

Reviewer #1: The authors have addressed all my concerns in the revised manuscript. I believe it to be acceptable for publication as it is now.

Reviewer #2: The authors conducted a comprehensive study of the effects of VDX-111, a vitamin D derivative, on a set of 30 well-characterized canine cell lines. The corrections made to the revised version of their manuscript have met my requests and in my opinion make the manuscript acceptable for publication.

7. PLOS authors have the option to publish the peer review history of their article (what does this mean?). If published, this will include your full peer review and any attached files.

Reviewer #1: No

Reviewer #2: **Yes: **Benjamin Cartiaux

---

## [Editor Report · Acceptance letter]

7 May 2024

PONE-D-24-03607R1 

PLOS ONE

Dear Dr. Farrell, 

I'm pleased to inform you that your manuscript has been deemed suitable for publication in PLOS ONE. Congratulations! Your manuscript is now being handed over to our production team.

Kind regards, 

on behalf of

Pr. Dominique Heymann 

Academic Editor

PLOS ONE